# Synonymous Codon Pair Recoding of the HIV-1 *env* Gene Affects Virus Replication Capacity

**DOI:** 10.3390/cells10071636

**Published:** 2021-06-29

**Authors:** Ana Jordan-Paiz, Sandra Franco, Miguel Angel Martinez

**Affiliations:** IrsiCaixa, Hospital Universitari Germans Trias i Pujol, Universitat Autònoma de Barcelona (UAB), 08916 Badalona, Spain; Ana.jordan@leibniz-hpi.de (A.J.-P.); sfranco@irsicaixa.es (S.F.)

**Keywords:** codon pair bias, synonymous gene recoding, HIV-1, envelope, fitness, attenuation

## Abstract

Synonymous codon pair deoptimization is an efficient strategy for virus attenuation; however, the underlying mechanism remains controversial. Here, we optimized and deoptimized the codon pair bias (CPB) of the human immunodeficiency virus type 1 (HIV-1) envelope (*env*) gene to investigate the influence of *env* synonymous CPB recoding on virus replication capacity, as well as the potential mechanism. We found that *env* CPB deoptimization did not always generate attenuation, whereas CPB optimization attenuated virus replication in MT-4 cells. Furthermore, virus attenuation correlated with reduced Env protein production but not with decreased viral RNA synthesis. Remarkably, in our model, increasing the number of CpG dinucleotides in the 5′ end of *env* did not reduce the replication capacity of HIV-1. These results indicate that factors other than CPB or CpG content may have impacted the viral fitness of the synonymously recoded study variants. Our findings provide evidence that CPB recoding-associated attenuation can affect translation efficiency. Moreover, we demonstrated that an increased number of CpGs in the 5′ end of HIV-1 *env* is not always associated with reduced virus replication capacity.

## 1. Introduction

Large-scale synonymous virus genome recoding has received much attention in recent years because it can be used to generate modified live attenuated vaccines [1,2]. The introduction of synonymous mutations in the protein coding region produces modifications in DNA or mRNA without altering the encoded proteins. Synonymous genome-wide recoding enables the synthetic generation of various small-genome viruses with modified phenotypes and biological properties. In the context of human immunodeficiency virus type 1 (HIV-1), synonymous recoding has elucidated new antiviral mechanisms within the innate immune response and improved our knowledge of new functional virus genome structures [3,4,5,6,7,8,9,10,11,12]. However, the mechanism underlying the alteration of virus phenotype through genome synonymous recoding has not yet been elucidated.

Synonymous recoding can change codon usage, codon pair usage, and dinucleotide frequencies (e.g., CpG or UpA), as previously reviewed [1,2]. Codon pair frequencies, termed codon pair bias (CPB), vary among different organisms. Any given genome exhibits different codon pair frequencies than would be randomly expected based on the individual codon usage bias of that genome. CPB was first described in bacteria [13] and was then detected in all studied organisms, including humans [14]. Increasing underrepresented codon pairs has led to the generation of highly attenuated viruses of several highly pathogenic human viruses, including poliovirus [14], influenza A virus [15], HIV-1 [10], respiratory syncytial virus (RSV) [16], and dengue virus [17]. It remains controversial whether CPB is shaped by selection. CPB deoptimization inadvertently increases the numbers of CpG and TpA (UpA) dinucleotides in recoded sequences [18,19] since codon pairs containing CpG dinucleotides at the codon pair boundary are among the most underrepresented in vertebrates. It has been hypothesized that increased CpG frequency is the sole cause of CPB-based attenuation.

HIV-1 mimics the CpG dinucleotide suppression of its human host [6]. Recent data suggest that the zinc-finger antiviral protein (ZAP) may be a driving force behind the CpG dinucleotide suppression in HIV-1 [11]. Cytoplasmic ZAP recognizes CpG dinucleotides in HIV-1 RNA, leading to its degradation and subsequent reductions of viral protein production and virus replication. Experimental introduction of CpGs into the HIV-1 genome significantly increased its susceptibility to ZAP [11], suggesting that HIV-1 evades ZAP restriction by reducing the number of CpGs within its genome. Nevertheless, it remains unclear whether ZAP-mediated restriction is the only mechanism driving CpG suppression. The anti-HIV-1 effect of CpGs does not correlate with CpG abundance [5], and insertion of CpGs into certain regions of the genome more efficiently sensitizes the virus to ZAP antiviral activity [5,7].

In the present study, we aimed to investigate the effects of HIV-1 envelope (*env*) codon pair recoding on virus replication capacity and Env protein expression. To this end, we codon pair optimized and deoptimized the HIV-1 *env* gene and examined how the replication capacity of recoded viruses was influenced by the contributions of underrepresented or overrepresented codon pairs versus CpG dinucleotides. To distinguish the effects of codon pair recoding and CpG frequencies, we designed and characterized multiple *env* mutants in which these two parameters were independently modified. Our results confirmed that codon pair recoding impacted protein production and virus replication capacity, independently of CpG frequencies.

## 2. Materials and Methods

### 2.1. Cell Lines

MT-4 cells were obtained from the NIH AIDS Reagent Program, Division of AIDS, NIAID, NIH. These cells were propagated in RPMI 1640 medium (Gibco, Thermo Fisher Scientific, Paisley, Scotland, United Kingdom) supplemented with 10% heat-inactivated fetal bovine serum (FBS) (Gibco, Thermo Fisher Scientific, Paisley, Scotland, United Kingdom). HEK 293T cells were obtained from the American Type Culture Collection (ATCC) and were grown in Dulbecco’s modified Eagle’s medium (DMEM) (Gibco, Thermo Fisher Scientific, Paisley, Scotland, United Kingdom) supplemented with 10% heat-inactivated FBS.

### 2.2. Generation of Synthetic Infectious HIV-1 Env Variants

The *env* sequences were designed based on the HXB2 strain (www.hiv.lanl.gov, accessed on 1 February 2010). To avoid the introduction of rare human codons, CPB-recoded *env* genes were designed with an improved CAI score. All sequences designed for this study were produced by Invitrogen GeneArt Gene Synthesis (Thermo Fisher Scientific, Waltham, MA, USA). To obtain infectious viral particles, MT-4 cells were electroporated with three overlapping PCR-amplified fragments, as previously described [8,20]. Briefly, full-length *env* was PCR amplified using an oligonucleotide pair: forward, 5′-GAGAGTGAAGGAGAAGTATCAGC-3′ and reverse, 5′-GCAAAATCCTTTCCAAGCCCTTGTC-3′. We also generated two PCR-amplified HIV-1 genome fragments. One extended from positions 1 to 6306 (primer pair: forward, 5′-TGGAAGGGCTAATTTGGTCC-3′ and reverse, 5′-CTACAGATCATCAATATCCC-3′. The other extended from positions 8369 to 9709 (primer pair: forward, 5′-ACCCACCTCCCAACCCCGAG-3′ and reverse, 5′-TGCTAGAGATTTTCCACACT-3′. These three purified PCR products (600 ng each) were co-transfected into MT-4 cells. At 4 and 7 days post-transfection, we quantified the HIV-1 p24 antigen using the Genscreen HIV-1 antigen (Ag) assay (Bio-Rad, Marnes-la-Coquette, France) to monitor viral production. Viral particles were collected when the HIV-1 p24 antigen concentration surpassed 500 ng/mL. If p24 was not detected after 7 days of culture, the cells were blind passaged with fresh medium to recover the virus. A designed variant was considered lethal when p24 was not detected by 30 days post-transfection. Virus titration was performed in MT-4 cells, and values were expressed in terms of the tissue culture dose for 50% infectivity (TCID50), as previously described [9].

### 2.3. Replication Capacity Assays

Viral replication capacity in MT-4 cells was determined as previously described [8,9,10]. Briefly, 0.8 × 10^6^ MT-4 cells were infected at a multiplicity of infection (MOI) of 0.001 (800 TCID50s) and incubated for 3 h at 37 °C and 5% CO_2_. Then, the cells were washed twice with phosphate-buffered saline (PBS) and resuspended in RPMI 1640 supplemented with 10% heat-inactivated FBS in a six-well plate. Every 24 h for 7 days, we collected 300 µL of culture supernatant and quantified the viral p24 antigen. Growth kinetics were determined by fitting a linear model to the log-transformed p24 data during the exponential growth phase with a maximum likelihood method, as previously described [8,9,10].

### 2.4. Plasmids and HIV-1 Env Expression

The *env* variants and WT *rev* were cloned in the pcDNA3.1D/V5-His-TOPO (Invitrogen, Thermo Fisher Scientific, Carlsbad, CA, USA) expression vector, as previously described [8]. Briefly, the WT and variant *env* sequences were cloned using an oligonucleotide pair: forward, 5′-CACCAAACAAGTAAGTATGAGAGTGAAGGA GAAATAT-3′ and reverse, 5′-TCATAGCAAAATCCTTTCCAAGCCC-3′. A splicing donor sequence was included upstream of the forward primer, as previously described [21]. The oligonucleotides used to clone *rev* were 5′-CACCATGGCAGGAAGAAGCGGA-3′ (forward) and 5′-TTCTTTAGCTCCTGACTCC-3′ (reverse). For plasmid transfection, 0.6 × 10^6^ HEK 293T cells were seeded into six-well plates 24 h before transfection. Then, the cells were transfected with 1.5 µg of WT or mutant Env expression plasmids and 0.5 µg of the WT Rev expression plasmid. Transfections were performed using Lipofectamine 3000 reagent (Invitrogen), and cells were collected 48 h after transfection.

### 2.5. Immunoblot Analyses and Antibodies

Immunoblotting was performed as previously described [8]. Briefly, 10^6^ HEK 293T cells were resuspended in 100 µL of cell extraction buffer (Thermo Fisher Scientific, Vienna, Austria) supplemented with phenylmethylsulfonyl fluoride and complete protease inhibitor cocktail (Sigma-Aldrich, Saint Louis, MO, USA). Cell lysates (10 µg protein) were separated by electrophoresis on NuPage 4% to 12% bis-Tris gels (Thermo Fisher Scientific, Waltham, MA, USA), and then blotted onto nitrocellulose membranes (Thermo Fisher Scientific, Waltham, MA, USA). Finally, these membranes were probed with antibodies against Hsp90 (C45G5; Cell Signaling Technology, Beverly, MA, USA), gp120 (anti-gp120 HIV-1 polyclonal antibody; American Research Products, Waltham, MA, USA), and V5 (V5 tag monoclonal antibody, for Rev detection; Invitrogen, Thermo Fisher Scientific, Carlsbad, CA, USA).

### 2.6. Real-Time Quantitative PCR

To quantify mRNA levels, we collected two vials of 10^6^ HEK 293T cells from each culture that had been transfected with an expression plasmid. Total RNA was extracted using the High Pure RNA isolation kit (Roche LifeScience) [8]. From 500 ng RNA, cDNA was synthesized using the PrimeScript RT master mix (Perfect Real Time; TaKaRa, Kyoto, Japan). The obtained transcripts were used to conduct quantitative PCR (qPCR) with TaqMan universal master mix (Applied Biosystems, Thermo Fisher Scientific, Carlsbad, CA, USA). We amplified Env mRNAs using oligonucleotides that targeted the RRE region (forward, 5′-CAGTGGAATAGGAGCTTTGT-3′ and reverse, 5′-TGTACCGTCAGCGTCATTG-3′) and a 6-carboxyfluorescein (FAM)-5′-CTTGGGAGCAGCAGGAAGCACTAT-3′ reporter. To quantify Rev mRNA, we used the oligonucleotides 5′-GCCCGAAGGAATAGAAGAAGAA-3′ (forward), 5′-GATCGTCCCAGATAAGTGCTAAG-3′ (reverse), and FAM-5′-TGGAGAGAGAGACAGAGACAGATCCA-3′ (reporter). As an internal or endogenous control, we used the glyceraldehyde-3-phosphate dehydrogenase (GAPDH) TaqMan gene expression assay (Hs99999905_m1; Applied Biosystems, Thermo Fisher Scientific, Carlsbad, CA, USA). The number of RNA copies per microgram of total cytoplasmic and nuclear purified RNAs was calculated using the corresponding plasmid DNA as a standard.

### 2.7. Statistical Analysis

To determine the significance of the differences between replication kinetic slopes, we performed an unpaired *t* test and Welch’s correction, as implemented in GraphPad Prism 8.3.0.

## 3. Results

To evaluate the effects of CPB on protein expression and virus viability, we designed and constructed a series of HIV-1 variants carrying CPB-recoded mutants of the HIV-1 *env* gene. CPB can be quantified using codon pair score (CPS) statistics [14]. The CPS of a gene is the arithmetic mean of individual CPS values, with negative CPS values indicating underrepresented codon pairs and positive CPS values indicating overrepresented codon pairs. We first constructed three entire CPB-recoded *env* genes: an optimized CPB variant (Max; increased CPS), a deoptimized CPB variant (Min; decreased CPS), and a neutral CPB variant (Neu; average CPS) (Table 1). The *Env* Max, Min, and Neu variants, respectively, contained 361 (13.9% of *env*), 427 (16.4%), and 396 (15.2%) mutations (Appendix A; Table 1). To avoid altering other viral genes, we did not modify regions that overlapped other reading frames. Similarly, we did not mutate the Rev response element (RRE), which is essential for virus viability. The three variants each showed an increased codon adaptation index (CAI), a measure of codon bias, indicating that rare human codons were not introduced (Table 1). When HIV-1 infectious clones containing the variants were transfected into susceptible MT-4 cells, we detected no virus growth or HIV-1 p24 antigen at 30 days post-transfection. Five blind passages of the transfected cell cultures yielded either virus or p24 antigen production.

Following CPB recoding, the introduction of CpG dinucleotides is almost inevitable because the most underrepresented codon pairs contain CpG dinucleotides at the codon pair boundary (NNC-p-GNN) [1]. Increasing the number of CpGs in HIV-1 *env* reportedly compromises virus viability and survival by increasing virus sensitivity to ZAP [11]. Indeed, our constructed *env* genes exhibited a significantly increased number of CpGs (Table 1). Therefore, we designed and constructed four new *env* variants—MaxCpG, MinCpG, MinCpG.2, and NeuCpG—which contained reduced numbers of CpGs (Appendix A; Table 1). The MinCpG.2 variant was designed in an effort to obtain a deoptimized variant with a lower CPS. However, although MinCpG and MinCpG.2 have different sequences, they exhibit very similar CPS values (Table 1). For all new variants, some CpGs were reverted to WT, and others were mutated anew, but the CPS values remained almost identical to those of the previous variants (Table 1). Again, CAI values were also increased to avoid rare codons. While the MaxCpG, MinCpG, and NeuCpG variants did not yield a viable virus, MinCpG.2 produced a viable virus that was capable of replication. Sequencing of the recovered MinCpG.2 confirmed the designed sequence. Replication kinetics experiments in MT-4 cells demonstrated that the replication kinetics were indistinguishable between the MinCpG.2 variant and WT virus (Figure 1A,B).

We previously demonstrated that virus replication and Env expression were completely abolished by synonymously changing a single codon (AGG) in the 3′ part of *env*, in the glycoprotein (gp)41 coding region (HXB2 *env* position 2125 to 2127), which is included in the intronic splicing silencer (ISS) [8]. Computational analysis of *env* sequences carrying mutations in this codon reveals severe disruption of the ISS RNA secondary structure [8]. Among the six variants tested in our present study, all except the MinCpG.2 variant have mutations in the above-mentioned codon. This result strongly suggests that gp41 synonymous recoding influenced the replication capacity of the generated variants. Thus, we next generated six new variants by modifying the previously tested variants to revert the gp41 coding region to WT (Appendix A). These six new variants exhibited CPS and CAI values comparable to those of the previous tested mutants (Table 1). After MT-4 cells were transfected with these six new mutants, all cell cultures exhibited HIV-1 p24 antigen production. However, the Min-3′WT variant did not produce a viable replication-competent virus. HIV-1 RNA sequencing of the new mutants confirmed the presence of the designed variants in the supernatant of all transfected MT-4 cell cultures.

The replication kinetics of Neu-3′WT, NeuCpG-3′WT, and MinCpG-3′WT were indistinguishable from those of the WT virus (Figure 2A,B). In contrast, Max-3′WT and MaxCpG-3′WT replicated with significantly lower kinetics. These unexpected findings demonstrated that CPB deoptimization was not always accompanied by virus attenuation. Moreover, CPB optimization was accompanied by a highly significant reduction in virus replication capacity. Interestingly, Max-3′WT and MaxCpG-3′WT contain fewer mutations than Neu-3′WT, NeuCpG-3′WT, and MinCpG-3′WT (Table 1). Furthermore, attenuation was not due to an increased number of CpGs in this 5′ end of *env* (gp120) since MaxCpG-3′WT, which lacks the newly introduced CpGs (Table 2; Appendix A), exhibited significantly worse replication than Max-3′WT (Figure 2). Neu-3′WT, which contains a significantly higher number of CpGs in this gp120 coding region than the WT virus and MaxCpG-3′WT (Table 2), replicated with kinetics similar to the WT virus. These results demonstrated that the number of CpGs in the gp120 coding region did not correlate with the observed HIV-1 attenuation. It has been previously shown that the number of CpGs among the first 700 bases at the *env* 5′ end determines the ZAP sensitivity of HIV-1 and thus impacts virus viability [7]. Among these 700 nucleotides, our WT virus contains 4 CpGs, the Neu-3′WT variant contains 5 CpGs, and the Min-3′WT has 23 CpGs. Thus, the number of CpGs in this region of *env* may explain the lack of viability of Min-3′WT but does not explain the low fitness of Max-3′WT (4 CpGs) and MaxCpG-3′WT (2 CpGs) (Appendix A).

To understand the mechanism through which synonymous CPB recoding impacted viral replication, we cloned all of the generated *env* mutants carrying a WT sequence in the 3′ end of *env* (gp41) into a eukaryotic expression vector and co-transfected HEK 293T cells with this vector and an HIV-1 Rev expression vector. Immunoblots of the different variants indicated drastically reduced expression levels with the mutant proteins Max-3′WT, MaxCpG-3′WT, and Min-3′WT (Figure 3A). Overall, these results were consistent with those obtained with infectious viruses, with the lower protein expression correlating with the virus replication kinetics. We also quantified mRNA expression, revealing that the amounts of total virus RNA did not significantly differ among cells transfected with the different variants (Figure 3B). Overall, our experiments demonstrated that CPB synonymous recoding can have a substantial impact on HIV-1 Env protein expression.

## 4. Discussion

Codon pair deoptimization has been widely used to generate attenuated viruses [1,2,14], including HIV-1 [6,9,10]. We previously demonstrated that codon pair deoptimization of the HIV-1 *gag* and *pol* genes strongly compromises virus replication capacity and viability [9,10]. Codon pair deoptimization unintentionally increases the frequency of CpG dinucleotides in recoded sequences, and it has been hypothesized that this increased frequency of CpGs is responsible for the attenuation of mammalian codon pair-deoptimized viruses [18,19]. However, a recent study of the influenza virus suggests that underrepresented codon pairs, rather than CpG dinucleotides, are the primary determinants of virus attenuation following CPB deoptimization [22]. It has also been shown that codon pair bias can be used to increase mRNA stability and protein production. In the present study, we tested the extent to which *env* gene CPB recoding affected HIV-1 replication capacity and protein expression.

We found that codon pair deoptimization did not always attenuate virus replication in tissue culture. Intriguingly, increasing the number of human overrepresented codon pairs in HIV-1 gp120 also led to virus attenuation. Although codon pair deoptimization typically results in virus attenuation, previous work with RSV has also shown that codon pair optimization can affect virus gene expression, protein synthesis, and virus replication [23]. Importantly, virus attenuation was not associated with an increased CpG dinucleotide frequency in the recoded coding region. An increased number of CpGs in the codon pair-deoptimized variant did not alter HIV-1 replication capacity. Moreover, in a codon pair-optimized variant, virus replication capacity was hampered by a reduction in CpGs. In our model, HIV-1 attenuation by codon pair recoding was associated with reduced translation efficiency and Env protein production.

Many vertebrate RNA viruses exhibit selection against CpG dinucleotides. HIV-1 is no exception, with its genome displaying a significantly lower than expected number of CpGs [24]. Several studies demonstrate that increasing CpG frequencies have negative effects on HIV-1 fitness, as previously reviewed [6]. HIV-1 *env* CpG dinucleotide suppression reportedly enables the virus to evade inhibition by ZAP in MT-4 cells [11]. ZAP binds to regions of HIV-1 mRNAs that have high CpG content, thereby targeting them for degradation and reducing viral protein expression and replication [3,11,25,26]. Thus, evasion of ZAP restriction may explain HIV-1 CpG suppression. However, in our present study, the Neu-3′WT variant with an increased number of CpGs in gp120 (29 vs. 8) did not exhibit reduced replication capacity in MT-4 cells. It has been postulated that the susceptibility of HIV-1 to ZAP may be regulated by the CpG frequency in a specific part of *env* rather than the overall virus genomic CpG content [5,7]. In particular, the number of CpG dinucleotides within the first 700 bases at the 5′ end of the *env* gene reportedly determines the ZAP sensitivity of primary HIV-1 strains [7]. Most HIV-1 strains have between three and six CpGs in this region [7]. These findings may explain the high replication capacity of Neu-3′WT, which has only five CpGs in this region, and the lack of viability of Min-3′WT, which has 23 CpGs. However, it remains unclear why we also observed low replication capacity in Max-3′WT and MaxCpG-3′WT—which have four and two CpGs, respectively, among the first 700 nucleotides of gp120.

One potential explanation for the low-fit phenotype of Max-3′WT and MaxCpG-3′WT could be the possible modification of cis-acting RNA elements that regulate HIV-1 splicing. However, thoughtful synonymous recoding of the entire HIV-1 *env* gene has revealed near-normal splicing [4]. Indeed, our Neu-3′WT-, NeuCpG-3′WT-, and MinCpG-3′WT-recoded variants displayed WT replication kinetics. Previous research has identified the interferon-stimulated protein Schlafen 11 (SLFN11) as a factor enabling the innate immune system to selectively inhibit HIV-1 protein synthesis in a manner dependent on codon usage [12]. Interestingly, this effect of SLFN11 was reduced by human codon optimization of the HIV-1 genome [12]. We can speculate that altering the CPB led to differential modification of the codon usage of our mutants. Since all of the presently generated variants have analogous CAI values, it is unlikely that SLFN11 was responsible for the reduced protein expression and the consequently reduced viral fitness of Max-3′WT and MaxCpG-3′WT. Nevertheless, it cannot be completely discarded that dissimilar codon bias among recoded *env* variants could impact virus replication capacity. Similarly, it cannot be proven that UpA dinucleotide changes in recoded sequences are responsible for the different phenotypes of the variants [19]. Among our presently analyzed variants, the number of UpA dinucleotides in their genomes did not correlate with their replication levels.

As we previously showed [8], our present findings demonstrated that the 3′ end of the HIV-1 *env* (gp41) nucleotide sequence was highly sensitive to synonymous recoding. Prior reports show that the nucleotide conservation of gp41 is related to RNA secondary structure and innate cell response [27,28]. Our present findings corroborate our previous results [8], and the lethality observed in the gp41-mutated variants was likely due to disruption of the ISS RNA secondary structure.

To investigate the mechanism underlying the virus attenuation due to codon pair-based synonymous recoding, our *env* variants were individually expressed in a eukaryotic expression model. We found that the *env* variants associated with low virus fitness yielded a drastic decrease in protein production. This reduction in Env protein production was not associated with reduced virus RNA production. The presently used expression system does not require splicing and, therefore, the observed reductions in protein translation were not generated by splicing perturbations. These findings are in agreement with previous studies of codon pair-mediated virus attenuation [1,2,14]. Remarkably, our data indicate that the increased CpG dinucleotides in codon pair-deoptimized sequences had only minor effects on protein production. We propose that there may be multiple different mechanisms responsible for codon pair-based virus attenuation.

Overall, our present findings showed that codon pair synonymous deoptimization is not always deleterious for HIV-1 replication. Although codon pair deoptimization has been mainly used for virus attenuation, our results expand the utility of this approach for other biotechnological applications—for example, to attenuate a vector sequence. This method has the potential to add synergistic mechanisms to other approaches to sequence optimization or deoptimization. Our study was limited by the analysis of only a few variants. Further work is needed to clarify the mechanism through which codon pair optimization, which does not increase the number of CpG dinucleotides, can impair protein production and produce virus attenuation.

## Figures and Tables

**Figure 1 cells-10-01636-f001:**
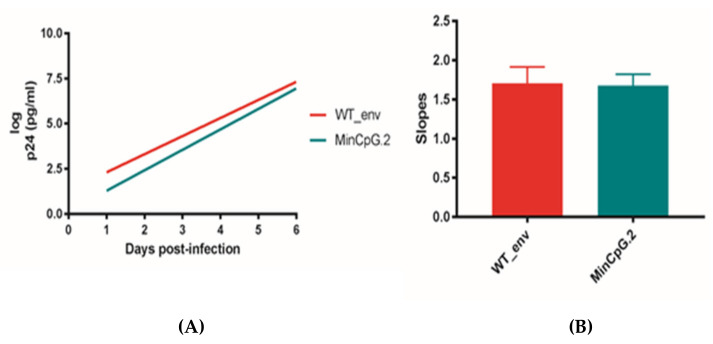
Replication kinetics of the CPB-recoded HIV-1 *env* MinCpG.2 variant in MT-4 cells compared to WT HXB2. To monitor viral replication, HIV-1 antigen p24 production was assessed in culture supernatants on days 0–6. (**A**) The slope of the p24 production plot provides an estimate of the viral replication capacity. Bars indicate this slope for the WT and MinCpG.2 variant viruses after infecting MT-4 cells. (**B**) The significance of the difference between slopes was calculated using GraphPad Prism 8.3.0. Values represent the mean ± standard deviation (SD) from at least three independent experiments.

**Figure 2 cells-10-01636-f002:**
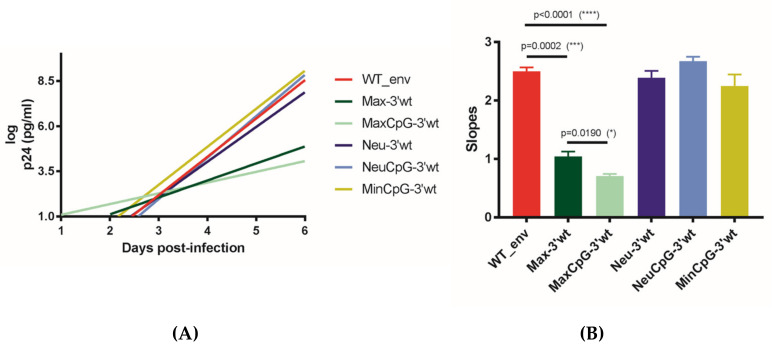
Replication kinetics of CPB-recoded HIV-1 *env* variants with the WT sequence in the *env* 3′ half (gp41) in MT-4 cells. To monitor viral replication, HIV-1 antigen p24 production was measured in culture supernatants on days 0–6. (**A**) The slope of the p24 production plot provides an estimate of the viral replication capacity. (**B**) Bars indicate this slope for WT and the *env* variant viruses after infecting MT-4 cells. The significance of the difference between the p24 antigen production slopes was calculated using GraphPad Prism 8.3.0. Values represent the mean ± SD from at least three independent experiments. One * means a *p* value of ≤ 0.05; *** *p* ≤ 0.001; **** *p* ≤ 0.0001.

**Figure 3 cells-10-01636-f003:**
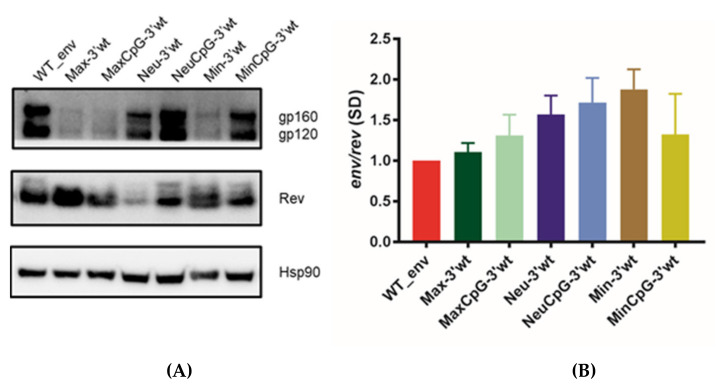
Env protein and mRNA production of CPB-recoded HIV-1 *env* variants with the WT sequence in the *env* 3′ half (gp41) in HEK 293T cells. Full-length WT and *env* variants were cloned into the pcDNA3.1D/V5-His-TOPO expression vector. Transfected cells were harvested after 48 h, and proteins were analyzed by SDS/PAGE and immunoblotting. (**A**) Immunoblot analyses of protein expression. The two bands in Env correspond to gp160 (upper) and gp120 (lower). The two bands in Rev correspond to the two isoforms. (**B**) Quantitative PCR results show total levels of Env mRNA. All samples were normalized to the WT mRNA level. Values represent the mean ± SD from at least three independent experiments.

**Table 1 cells-10-01636-t001:** Number of mutations, CpGs, UpAs, codon pair scores (CPS), and codon adaptation index (CAI) of the examined HIV-1 *env* variants.

Env	No. of Mutations	CpGs	UpAs	CPS	CAI
WT	-	26	194	0.046	0.137
Max	361	43	152	0.176	0.158
Neu	396	50	156	0.035	0.183
Min	427	111	144	−0.160	0.213
MaxCpG	348	21	155	0.160	0.156
NeuCpG	380	29	159	0.041	0.180
MinCpG	374	24	151	−0.085	0.189
MinCpG.2	354	23	158	−0.086	0.191
Max-3′WT	294	39	161	0.147	0.153
MaxCpG-3′WT	284	24	163	0.134	0.152
Neu-3′WT	319	47	164	0.030	0.173
NeuCpG-3′WT	305	30	166	0.035	0.170
Min-3′WT	350	103	156	−0.129	0.194
MinCpG-3′WT	297	26	163	−0.059	0.171

**Table 2 cells-10-01636-t002:** Number of CpGs located in the 5′ end of *env* (gp120) (first 1533 nucleotides of HXB2 *env*) in the variants carrying a WT sequence in the 3′ end of *env* (gp41).

Env	No. of CpGs	No. of UpAs
WT	8	128
Max-3′WT	21	95
MaxCpG-3′WT	6	97
Neu-3′WT	29	98
NeuCpG-3′WT	12	100
Min-3′WT	85	90
MinCpG-3′WT	8	97

## Data Availability

The data presented in this study are available on request from the corresponding author.

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
