# Peer review of "Synonymous Codon Pair Recoding of the HIV-1 env Gene Affects Virus Replication Capacity"

_cells, 2021, doi:10.3390/cells10071636_

Round 1

Reviewer 1 Report

Jordan-Paiz et al describe changes in virus replication capacity after synonymous codon pair recoding of the HIV-1 env gene. They show (i) that codon pair bias deoptimization not always lead to an attenuated phenotype, and (ii) that codon pair bias optimization can result in attenuation. The latter correlated with reduced  Env protein production  presumably as a result of reduced translation efficiency.

While the study is solely descriptive, it adds new information with respect to codon recoding and CpG content effects on virus replication. The results suggest translation regulation as an important factor of control. This is an area of intense investigation both in virology to understand codon bias effects as well as in basic science to decipher translation control mechanisms.

The experiments are well planned and sufficiently described.

Author Response

We acknowledge the reviewer comments.

Reviewer 2 Report

The manuscript by Ana Jordan-Paiz et al. describes the construction and analysis of HIV-1 envelope (env) mutants that were produced by synonymous codon recoding. Synonymous codon recoding, which includes also codon pair deoptimization, has been used to generate live attenuated viruses, such as the poliovirus, influenza virus, respiratory syncytial virus, dengue virus and Zika virus.

The authors constructed several codon pair optimized (Max), deoptimized (Min), and neutral (Neu) env gene variants, recovered infectious mutant viruses, and analyzed their replication in cell culture. In addition, they analyzed RNA and protein production of recoded genes in transiently transfected cells by RT-qPCR and Western blotting assays.

The major finding of this manuscript is that, contrary to expectations, two different codon pair-optimized env genes (Max-3'WT and MaxCpG-3'WT) produced less protein from the and mutants carrying these genes were attenuated in cell culture. Because both gene mutants have a higher average codon pair bias (CPB) score than the parental gene but also a relatively low number of CpG dinucleotides, factors other than CPB and CpG content must be responsible for the attenuation of these 2 mutants. However, the cause of the attenuation of these two mutants was not determined.

Major comments

Recently, the authors have showed that a codon AGG encoding arginine located at a specific position in the gp41 coding region (HXB2 env position 2125 to 2127) cannot be exchanged with a synonymous codon CGU, because this exchange disrupts the intronic splicing silencer, which constitutes a lethal mutation.

The authors generated six different env mutants (Max, Neu, Min, MaxCpG, NeuCpG, and MinCpG) that contain this lethal mutation. Oddly, the first sections of the Results describe recovery attempts of Max, Neu and Min, as well as MaxCpG, NeuCpG, MinCpG and MinCpG.2 virus mutants, as if the authors were not aware if the presence of this lethal mutation in the recoded sequences, and thus also futility of most of these (6/7) recovery attempts.

Perhaps these mutants were constructed and tested before the effect of the lethal mutation was discovered. In any case, I think it would be more appropriate to present the results at the outset in the light of current knowledge.

Perhaps it would even be more appropriate to remove the description of the failed recovery attempts of the Max, Neu, Min, MaxCpG, NeuCpG, and MinCpG mutants altogether, as it adds no value to the current work.

Although the authors state that they modify only CPB in recoded sequences, the sequence analysis clearly shows that sequences also have dissimilar codon bias, which authors also acknowledge in the discussion (lines 313-314). The fact that sequences differ not only in CPB but also in codon bias (codon usage) should be more clearly appreciated in the manuscript.

The authors claim in the abstract that env CPB deoptimization did led to virus attenuation, which contradicts the information shown in the Results – “the Min-3’WT variant did not produce a viable replication-competent virus” (lines 208-209) and in the Discussion – “These findings may explain … the lack of viability of Min-3’WT, which has 23 CpGs.” (lines 300-301).

Minor comments

The authors should provide full sequences of the recoded genes to allow better inspection of the recoding. They could be deposited in GenBank, or provided in the supplementary information.

The Min-3’WT sequence contains 103 CpG dinucleotides, not 130 as indicated in Table 1. In addition, most of the recoded sequences could also have a incorrect number of UpA dinucleotides.

The authors did not specify in either the M&M or Results sections how the recoded sequences were generated. They claim to have altered the codon pair bias (CPB) in the recoded sequences, but since the recoded sequences Max, Neu and Min have significantly different codon adaptation index (CAI) scores, it is clear that the recoded sequences also have considerably different codon bias. Why is this the case?

The statement about the inevitable introduction of CpG dinucleotides into sequences during CPB recoding is slightly imprecise (lines 170-172). In my opinion, the introduction of CpG dinucleotides into recoded sequences is inevitable because the most underrepresented codon pairs contain CpG dinucleotides at the codon pair boundary (NNC-p-GNN).

As expected, the env gene variant Max has a reduced number of UpA dinucleotides compared to the parental (WT) sequence, but the number of CpG dinucleotides is increased. Why does the Max sequence have an increased number of CpG dinucleotides?

The authors claim that "the MinCpG.2 variant was designed in an effort to obtain a deoptimized variant with a lower CPS" (line 178). Yet, both MinCpG and MinCpG.2 have almost identical CPS values. Is CPS of ~-0.085, the lowest achievable value?

Table 2 shows the number of CpG dinucleotides that are present in the 5’ end in some of the recoded env genes. However, the table does not specify exactly how long the 5’ end region was analyzed. Is this 5’ end proximal region 1,500 nt long?

The data shown in Table 2 are hardly discussed. Perhaps they could be replaced with data that show the number of CpG dinucleotides in the first 700 bases at the 5’ end of the env gene, which are mentioned in the manuscript.

I disagree with the statement that is present in the final summary of this manuscript indicting that results presented in this manuscript expanded the utility of codon pair deoptimization for other biotechnological applications – to improve protein production or vector sequence attenuation (lines 339-342). The manuscript shows no evidence that deoptimized genes would produce more protein.

Author Response

Please, find attached our point-by-point response.

Reviewer 3 Report

The authors codon optimized and deoptimized codon pair bias of HIV-1 env and found that its optimization attenuated the virus, whereas deoptimization had no such effect. This work involved extensive mutagenesis of the HIV envolope and yielded interesting results for virus attenuation strategies.

There are however certain issues that should be addressed:
Main issues:
1) It is unclear to me whether the used experimental set up results in production of full-length viral RNA containing introduced changes on the same message as the measured p24. In the context of HIV-1 env CpG targetting, ZAP has been shown to mainly target CpGs found in the unspliced or incompleteley spliced viral RNA, but not shorter CpG-rich messages such as reporter genes. This needs to be clarified, and if it is identified as a potential shortcoming of the method used, addressed in the discussion.
2) The conclusions of this paper rely on Gag p24 being a readout of viral production, but studied changes are introduced into env region only. Therefore it is important to check wheather produced virion-incorporated env levels (for example by western blot), virion RNA levels (qRT PCR) and infectivity (HIV-1 reporter cell assay, for example TZM-bl) can be affected to a higher extent than Gag, at least for the most important env mutants. 
3) Authors rely on replication curve slope calculations to draw conclusions from their data. This seems misleading in some cases, as for example in Fig 1, the WT virus produces more p24 at all sampled time points, and the difference is not minor, as the data is shown on a log scale. However the slope calculation on the right does not show any difference between the two viruses. Authors should either calculate mean viral p24 production or area under the replication curve and perform statistics on these for mutants tested in Fig. 1 and 2 to support their conclusions.

Minor issues:
3) On Fig 3, the rev levels used for normalization, especially for Neu-3'wt, do not look very even. What is the reason for this? Is this blot representative of what was observed in the other experiments? Such a pronounced difference in control protein levels could point to a potential mechanism of attenuation of some of the tested mutants.
4) Table 2 should list the number of UpAs as well as that of CpGs.

Author Response

(The authors gave the same response as above.)

Round 2

Reviewer 2 Report

The authors addressed most of my concerns and corrected all the errors I found initially.

I still believe that the first part of the results section should not contain the misleading information. However, this is only a minor issue that does not compromise the overall quality of the study presented. 

Reviewer 3 Report

I have no further comments.